# Observation of the exceptional point in cavity magnon-polaritons

Dengke Zhang[1,3], Xiao-Qing Luo[1], Yi-Pu Wang[1], Tie-Fu Li[2] & J.Q. You[1]

Magnon–polaritons are hybrid light–matter quasiparticles originating from the strong coupling between magnons and photons. They have emerged as a potential candidate for implementing quantum transducers and memories. Owing to the dampings of both photons and magnons, the polaritons have limited lifetimes. However, stationary magnon–polariton states can be reached by a dynamical balance between pumping and losses, so the intrinsically nonequilibrium system may be described by a non-Hermitian Hamiltonian. Here we design a tunable cavity quantum electrodynamics system with a small ferromagnetic sphere in a microwave cavity and engineer the dissipations of photons and magnons to create cavity magnon–polaritons which have non-Hermitian spectral degeneracies. By tuning the magnon–photon coupling strength, we observe the polaritonic coherent perfect absorption and demonstrate the phase transition at the exceptional point. Our experiment offers a novel macroscopic quantum platform to explore the non-Hermitian physics of the cavity magnon–polaritons.

[1] Quantum Physics and Quantum Information Division, Beijing Computational Science Research Center, Beijing 100193, China. [2] Institute of Microelectronics, Tsinghua National Laboratory of Information Science and Technology, Tsinghua University, Beijing 100084, China. [3]Present address: Department of Engineering, University of Cambridge, Cambridge CB3 0FA, UK. Correspondence and requests for materials should be addressed to T.-F.L. (email: litf@tsinghua.edu.cn) or to J.Q.Y. (email: jqyou@csrc.ac.cn)

Controlling light–matter interactions has persistently been pursued and is now actively explored[1]. Understanding these interactions is not only of fundamental importance but also of interest for various applications[2]. Recently, there has been an increasing number of studies on collective excitations of ferromagnetic spin system (i.e., magnons) coupled to microwave photons in a cavity (see, e.g., refs. [3–11]). Owing to the strong coupling between magnons and cavity photons, a new type of bosonic quasiparticles called cavity magnon–polaritons can be created. These cavity magnon–polaritons have short lifetimes due to the dissipative losses of photons and magnons, therefore requiring continuous pumping to compensate dissipations, so as to reach a steady-state regime[5–8]. While the damping rate of magnons is fixed, by engineering the ports of cavity for inputting microwave photons and the related decay rates, the system of cavity photons coupled to magnons can be described by a non-Hermitian Hamiltonian. The hallmark of a non-Hermitian system is the existence of a singularity in its eigenvalues and eigenfunctions at some particular points in the parameter space of the system. This singularity is called the exceptional point[12, 13]. At this point, two separate eigenmodes coalesce to one. Distinct phenomena originating from the exceptional-point singularity

have been observed in various systems related to electromagnetism[14–16], atomic and molecular physics[17], quantum phase transitions[18, 19], quantum chaos[20], etc.

Recent work on magnon–photon interaction has explored the strong and even ultra-strong couplings of microwave cavity photons to the ferromagnetic magnons in yttrium iron garnet (YIG)[5–8, 21, 22]. This interaction enabled the coupling of magnons to a qubit[23] or phonons[24], microwave photons to optical photons[25–28], and multiple magnets to one another[29]. However, as an inherent non-Hermitian system, the related interesting properties have not been explored. Rather than a drawback of the system, the intrinsic nonequilibrium character enriches the phenomena related to the cavity magnon–polaritons.

Here we experimentally demonstrate that the non-Hermiticity dramatically modifies the mode hybridization and spectral degeneracies in cavity magnon–polaritons. In our experiment, we engineer the dissipations of magnons and photons to produce an effective non-Hermitian PT-symmetric Hamiltonian. By tuning the magnon–photon coupling, we observe the polaritonic coherent perfect absorption (CPA) and demonstrate the phase transition at the exceptional point. Thus, cavity magnon–polaritons with non-Hermitian nature are explored

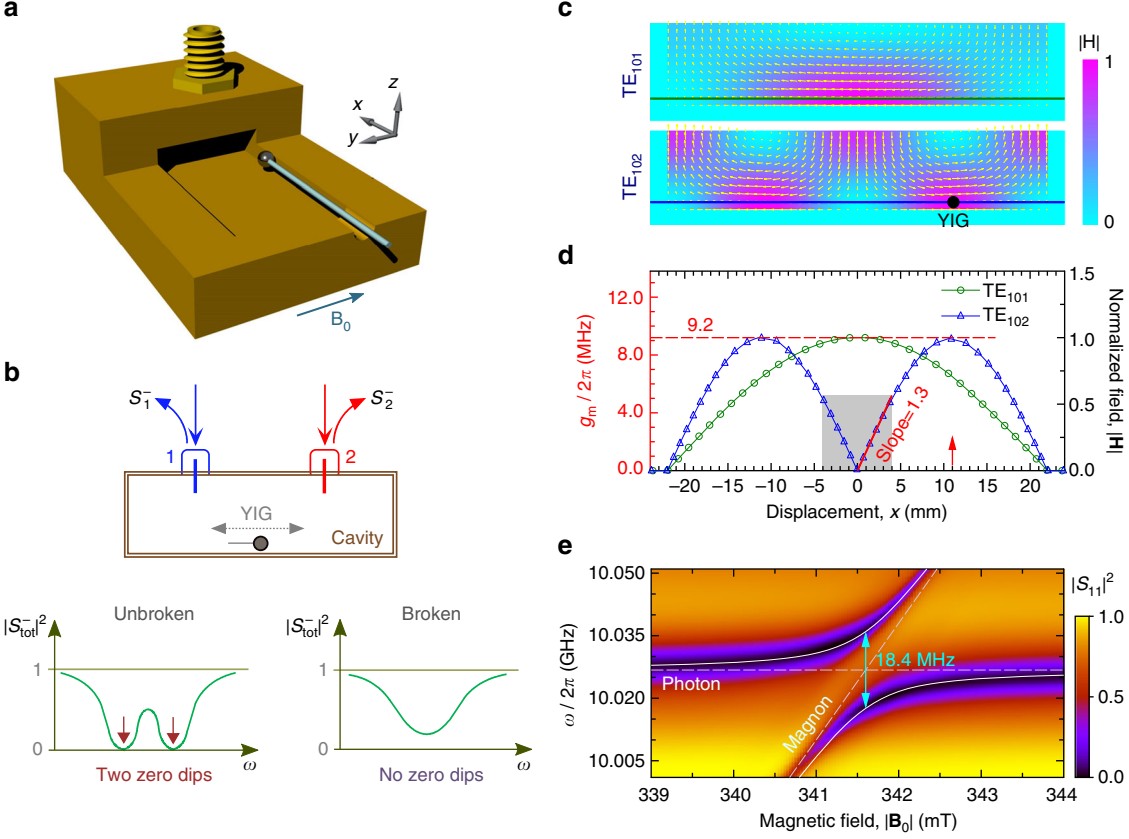

**Fig. 1** Cavity magnon–polaritons. **a** Sketched structure of the cavity magnon–polariton system, where a YIG sphere glued on a wooden rod is inserted into a 3D rectangular cavity through a hole of 5 mm in diameter in one side of the cavity. The displacement of the YIG sphere can be adjusted along the x direction using a position adjustment stage and the static magnetic field is applied along the y direction. The cavity has two ports for both measurement and feeding microwave fields into the cavity. **b** Diagram of the cavity magnon–polariton system with two feedings. The total output spectrum $|S^-_{tot}|^2$ is the sum of output spectrum $|S^-_1|^2$ from port 1 and $|S^-_2|^2$ from port 2. When the system is designed to possess PT symmetry as described by Eq. (2), there is an exceptional point. In the unbroken-symmetry regime, the total output spectrum has two coherent perfect absorption (CPA) frequencies, but no CPA occurs in the broken-symmetry regime. **c** Normalized microwave magnetic-field distributions of the cavity modes $TE_{101}$ and $TE_{102}$. **d** Variations of the microwave magnetic fields of the $TE_{101}$ and $TE_{102}$ modes along the moving path of the YIG sphere marked in **c**. The gray area corresponds to our experimental region, where the magnetic-field intensity of the $TE_{102}$ mode shows an approximately linear relation with the displacement of the YIG sphere and the varying slope is estimated to be 1.3 MHz/mm. **e** The maximal coupling of the $TE_{102}$ mode to magnon is reached at the displacement $|x| \sim 11$ mm of the YIG sphere, with the fitted coupling strength given as 9.2 MHz

and achieved in our experiment. It paves the way to explore the non-Hermitian physics of the cavity magnon–polaritons.

## Results

**Cavity magnon–polaritons.** In our device, a small YIG sphere with a diameter of 0.32 mm is embedded in a three-dimensional (3D) rectangular microwave cavity with two ports (Fig. 1a, b). This rectangular cavity is produced using oxygen-free high-conductivity copper and has dimensions $44 \times 20 \times 6$ mm$^3$. The YIG sphere is glued on a thin wooden rod (about 1 mm in diameter) and inserted into the cavity from a side hole of 5 mm in diameter. By tuning the insertion depth of the rod with a position adjustment stage, the YIG sphere is moved along the long edge of the rectangular cavity. To tune the frequency of the magnon in the YIG sphere, the device is placed in a static magnetic field ($\mathbf{B_0}$) whose direction is parallel to the short edge of the cavity. The two lowest-order resonance modes of the cavity are TE$_{101}$ and TE$_{102}$ (Fig. 1c, d). Initially, we place the YIG sphere at the site where the microwave magnetic field of the TE$_{102}$ mode takes the maximum value.

Here we focus on the case with magnons resonantly interacting with the cavity mode TE$_{102}$. At low-lying excitations, the system can be described by the Hamiltonian[3–8] (when adopting the rotating-wave approximation and setting $\hbar = 1$), $H_s = \omega_c a^\dagger a + \omega_m b^\dagger b + g_m(ab^\dagger + a^\dagger b)$, where $a^\dagger(a)$ is the creation (annihilation) operator of microwave photon at frequency $\omega_c$, $b^\dagger(b)$ is the creation (annihilation) operator of magnon at frequency $\omega_m$, and $g_m$ is the magnon–photon coupling strength. The total field damping rate $\kappa_c$ of the cavity mode TE$_{102}$ includes the decay rates $\kappa_i$ ($i = 1, 2$) induced by the two ports and the intrinsic loss rate $\kappa_{int}$ of the cavity mode, i.e., $\kappa_c = \kappa_1 + \kappa_2 + \kappa_{int}$. The damping rate $\gamma_m$ of the magnon comes from the surface roughness as well as the impurities and defects in the YIG sphere[5, 8]. When the magnon frequency is close to the photon frequency, the strong interaction between magnon and cavity photon mixes each degree of freedom and creates the hybridized states of the cavity magnon–polaritons.

To get the spectrum of the system, the microwave signal is injected into the cavity via port 1, reflected by the device, and detected using a vector network analyzer (VNA) via a 20 dB directional coupler (Supplementary Note 1). By sweeping the static magnetic field, the reflection spectrum of the port 1 of the cavity is recorded and plotted in Fig. 1e. It is clear that there exists strong coupling (reflected by an apparent anti-crossing) between microwave photons and magnons. Here the considered magnon mode is the spatially uniform mode (i.e., the Kittel mode), whose frequency depends linearly on the static magnetic field, $\omega_m = \gamma_e|\mathbf{B_0}| + \omega_{m,ai}$, where $\gamma_e$ is the electron gyromagnetic ratio and $\omega_{m,ai}$ is determined by the anisotropic field[6]. Using input–output theory[30], we can calculate the reflection spectrum $S_{11}(\omega)$. Then, by fitting with the experimental results, we obtain $g_m/2\pi = 9.2$ MHz for the coupling of the Kittel mode with the cavity mode TE$_{102}$ and $\gamma_m/2\pi = 1.5$ MHz for the magnon.

**Polaritonic CPA.** In a dissipative quantum system, a steady state requires persistent pumping to continuously compensate the dissipations. A distinct phenomenon in a cavity polariton system is the polaritonic CPA which occurs at a critical coupling between the input channel and the load[31, 32]. For our system, there are two feedings at both the two ports (Fig. 1b). When the CPA occurs, all the outgoing fields ($S_1^-$ and $S_2^-$) from the two feeding ports become zero and the system can be effectively described by a non-Hermitian Hamiltonian (Supplementary Note 2),

$$H_{CPA} = [\omega_c + i(\kappa_1 + \kappa_2 - \kappa_{int})]a^\dagger a + (\omega_m - i\gamma_m)b^\dagger b + g_m(ab^\dagger + a^\dagger b). \quad (1)$$

Here, without using a gain material, we harness the feeding fields to achieve an effective gain in Eq. (1). The eigenfrequencies of this effective Hamiltonian are usually complex and one cannot have direct experimental observation in this complex-frequency case. However, if $\omega_c = \omega_m = \omega_0$ and $\kappa_1 + \kappa_2 - \kappa_{int} = \gamma_m$ are satisfied by tuning the system parameters, the effective non-Hermitian Hamiltonian is reduced to

$$H_{CPA} = (\omega_0 + i\gamma_m)a^\dagger a + (\omega_0 - i\gamma_m)b^\dagger b + g_m(ab^\dagger + a^\dagger b). \quad (2)$$

It can be found that the Hamiltonian in Eq. (2) satisfies $[PT, H_{CPA}] = 0$, so the CPA in this system can be effectively described by the non-Hermitian PT-symmetric Hamiltonian[13, 33, 34]. The corresponding eigenvalues are solved as

$$\omega_{1,2} = \omega_0 \pm \sqrt{g_m^2 - \gamma_m^2}. \quad (3)$$

In Eq. (3), the two eigenfrequencies $\omega_{1,2}$ are functions of $\omega_0$, $\gamma_m$, and $g_m$, and referred to as the CPA frequencies. To have a real spectrum, it requires $g_m > \gamma_m$, and $\omega_{1,2}$ coalesce into the central frequency $\omega_0$ at $g_m = \gamma_m$. While $g_m < \gamma_m$, $\omega_{1,2}$ become complex; the symmetry of the system is spontaneously broken and the CPA then disappears (Fig. 1b). The exceptional point at $g_m = \gamma_m$ is also referred to as the spontaneous symmetry-breaking point. Meanwhile, it can be calculated that $\omega_{1,2}$ are exactly the zeros of the total outgoing spectrum $|S_{tot}^-|^2$ $\left(= |S_1^-|^2 + |S_2^-|^2\right)$, which can be achieved when phase difference of two feeding fields is set as $\Delta\phi = 0$ and the power ratio is confined to $q = \kappa_1/\kappa_2$ (Supplementary Note 2).

In order to achieve the effective Hamiltonian in Eq. (2) and then observe the CPA, it is required to achieve $\omega_c = \omega_m$ and $\kappa_1 + \kappa_2 - \kappa_{int} = \gamma_m$ in our system. In the experiment, the former requirement can easily be realized by tuning $\omega_m$ via $|\mathbf{B_0}|$. To meet the latter requirement, the dissipation rate of the cavity mode is tailored in our device. By tuning the pin length of each port inside the cavity and fitting the measured total damping rate, $\kappa_{int}/2\pi$ is estimated to be 1.55 MHz for the cavity mode. For the port-induced decay rates $\kappa_1$ and $\kappa_2$, their magnitudes can be tuned by adjusting the lengths of the port pins inside the cavity. In our experiment, $\kappa_1/2\pi$ and $\kappa_2/2\pi$ are tuned to be 1.72 and 1.39 MHz, respectively, for the bare cavity (Fig. 2a), which satisfy the required relation $\kappa_1 + \kappa_2 - \kappa_{int} = \gamma_m$ well (because $\gamma_m/2\pi = 1.5$ MHz in our YIG sample). It should be noted that when the YIG sphere is inserted into the cavity, both the cavity frequency and intrinsic loss rate shift slightly at different displacements of the YIG sphere (Supplementary Note 3). These tiny shifts can affect the observed results, but we can use them as modifications in the simulations, so as to make a comparison with the experimental results.

For the measurement setup with two feedings, the input microwave field is divided into two copies and injected into the cavity via the two ports. Then, we detect the outgoing microwave fields from both ports (Supplementary Note 1). As in the previous description, to achieve the CPA in the experiment, phase difference of the two feeding fields should be set as $\Delta\phi = 0$ and the power ratio should be confined to $q = \kappa_1/\kappa_2$, which equals 1.23 in our experiment. In order to tune the power ratio and the phase difference between feeding fields of ports 1 and 2, a variable attenuator and a phase shifter are used in the experiment. Then,

**a**

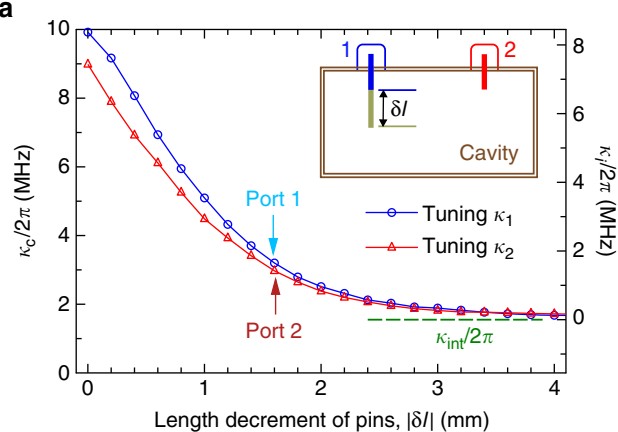

**b**

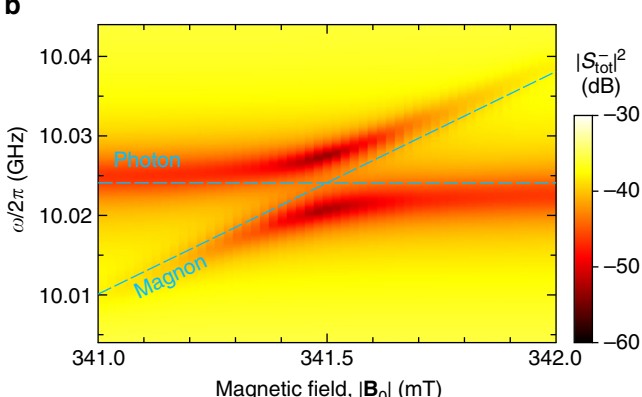

**Fig. 2** Decay rate of the cavity and the CPA. **a** Effects of the pin lengths of ports 1 and 2 on port-induced decay rates $\kappa_i$ ($i = 1$, 2) and the total decay rate $\kappa_c$ of the cavity mode. When the length decrements of the pins are adjusted to be large enough, $\kappa_c$ tends to a constant and the intrinsic loss rate $\kappa_{int}$ is then evaluated. At the data points marked by blue and red arrows for ports 1 and 2, we observe the CPA. Inset: The sketch for tuning the pin in a port of the cavity. **b** Total output spectrum measured at $x = -3$ mm by sweeping the static magnetic field at $\Delta\phi = 0$ and $q = 1.23$

to guarantee $g_m > \gamma_m$ for the CPA, we put the YIG sphere at $x = -3.0$ mm, where $g_m$ is about 3.9 MHz, as obtained from Fig. 1d. In the experiment, we monitor the outgoing fields from both port 1 $(S_1^-)$ and port 2 $(S_2^-)$ by sweeping the magnon frequency. As shown in Fig. 2b, when the magnon frequency approaches to the resonant frequency of the cavity, anti-crossing of these two frequencies occurs and two deep dips appear in the total output spectrum $\left|S_{tot}^-\right|^2$. It corresponds to the CPA in this two-port case.

**Observation of the exceptional point**. When the CPA is achieved, we can further tune the coupling strength to reach the exceptional point according to Eq. (3). There must be a tunable external parameter to observe the exceptional point and $g_m$ can be chosen in our experiment. Here, tuning $g_m$ can be realized by adjusting the displacement of the YIG sphere along the $x$ direction, because the coupling strength is proportional to the magnitude of the microwave magnetic field. It is measured that $g_m/2\pi$ varies approximately linearly with $|x|$ and the varying slope is about 1.3 MHz/mm in the region of $|x| \leq 4$ mm (Fig. 1d). Figure 3a illustrates the measured total output spectrum $\left|S_{tot}^-\right|^2$ at different displacements of the YIG sphere. It is clearly seen that when $|x|$ is larger than 1.2 mm, there exist two CPA frequencies (at very low output), corresponding to the two real eigenfrequencies of the PT-symmetric Hamiltonian; when $|x|$ is smaller

than 1.2 mm, a phase transition occurs and no CPA frequency can be found. Now the system exhibits an exceptional point differentiating the two regimes of unbroken and broken symmetry. Figure 3b shows the simulated $\left|S_{tot}^-\right|^2$, in good agreement with the experimental results, where the corresponding coupling strength is also marked according to the relation $g_m/2\pi = 1.3|x|$. In Fig. 3a, b, open-circle curves are the eigenfrequencies calculated using Eq. (3), which are also in good agreement with the CPA frequencies. It can be found that the coupling strength corresponding to the exceptional point is 1.5 MHz, as expected to be equal to the value of $\gamma_m/2\pi$.

To further display the difference between the output spectra of the system in both the unbroken- and broken-symmetry regimes, we demonstrate the dependence of the total output spectrum on $\Delta\phi$ and $q$. In Fig. 3c, we can see that, for the two CPA frequencies at $x = -3.0$ mm (marked as I and II in Fig. 3a), the CPA occurs at the optimal value of $\Delta\phi = 0$ and the variation of outgoing power is more than 20 dB with respect to the adjustment of $\Delta\phi$ at a fixed $q$ of 1.23. However, this adjustment has a weak influence on the output spectrum for the non-CPA frequency at $x = 0$ (marked as III in Fig. 3a), where the power variation is less than 10 dB. Similar phenomena are also observed by adjusting $q$ while keeping $\Delta\phi = 0$, as displayed in Fig. 3d. In short, when the system is in the unbroken-symmetry regime, the output spectrum of the system is highly sensitive to both $\Delta\phi$ and $q$ of the two feeding fields, but this sensitivity is significantly reduced when the system goes into the broken-symmetry regime.

## Discussion
According to the relative values among $g_m$, $\kappa_c$, and $\gamma_m$, there are four parameter regions with distinct phenomena, i.e., the regimes with strong coupling ($g_m > \kappa_c$, $\gamma_m$), magnetically induced transparency (MIT) ($\kappa_c > g_m > \gamma_m$), Purcell effect ($\gamma_m > g_m > \kappa_c$), and weak coupling ($g_m < \kappa_c$, $\gamma_m$)[6]. By tuning the system parameters, conversion between different regimes can be achieved. However, it is difficult to observe the accompanied phase transition, because a phase transition can appear only along a specific route. Note that $\kappa_c/2\pi = 4.66$ MHz and $\gamma_m/2\pi = 1.5$ MHz in our experiment. Thus, when $g_m/2\pi > 1.5$ MHz, the system lies in either strong-coupling ($g_m > \kappa_c$) or MIT ($g_m > \gamma_m$) regime, and there are two deep dips in the output spectrum, owing to the polaritonic CPA. When $g_m/2\pi < 1.5$ MHz, the system is in the weak-coupling regime ($g_m < \gamma_m$) and there is no apparent dip in the output spectrum. In Fig. 3e, we show the minima of $\left|S_{tot}^-\right|^2$ as a function of the displacement of the YIG sphere. Indeed, a phase transition is explicitly displayed at the displacement corresponding to the exceptional point. Therefore, the exceptional point observed above is also the phase transition point for the system to transition from the MIT regime to the weak-coupling regime.

Recently, cavity magnon–polaritons are proposed to be used as hybrid quantum systems for quantum information processing[23, 26], where the engineering of dissipation is a crucial part. In our experiment, we engineer the ports of the cavity and the related decay rates to implement a cavity magnon–polariton system with the non-Hermitian PT-symmetric Hamiltonian. We show that the CPA can be utilized to identify the transition from the unbroken-symmetry regime to the broken-symmetry regime at the exceptional points in this hybrid system. However, in optical systems, it has been shown[35] that self-dual spectral singularities and CPA can be obtained without PT symmetry. For the cavity magnon–polariton scheme, this should also be possible because the quantum description of the cavity magnon–polariton system under the low-lying excitation of magnons is analogous to an optical system. Moreover, as a new platform for non-Hermitian physics, the cavity magnon–polaritons possess the merits of both

magnons and cavity photons. It will provide more degrees of freedom to investigate possible designs stemming from the absence of PT symmetry, including the observation of exceptional points in the non-Hermitian system without PT symmetry[35–37].

In addition, magnons are intrinsically tunable and non-reciprocal. In contrast to the optical and atomic systems, the hybrid system of cavity magnon–polaritons can provide new possibilities in non-Hermitian physics. For instance, pronounced nonreciprocity and asymmetry in the sideband signals generated by magnon-induced Brillouin scattering of light were observed in such a system[27]. When non-Hermiticity is considered, these

properties may bring new features in the magnon–polariton system. Also, more small YIG spheres can be placed in a cavity. Tuning magnons in each YIG sphere, one can strongly couple magnons in these spheres to the cavity photons, either simulta-neously or respectively[29, 38, 39]. Then, one can implement quan-tum simulations of many-body bosonic models using this hybrid system and explore more interesting phenomena in non-Hermitian physics.

Moreover, nonreciprocal light transmission was observed in an optical system with PT symmetry[15]. With the similarity to the optical system, it seems possible to demonstrate nonreciprocal light transmission in the cavity magnon–polariton system. For this, one can use one port of the cavity and then design a special port for the YIG sphere by, e.g., directly coupling the microwave field to the magnons via a coil[40]. Therefore, a round-trip signal transmission path between the cavity and the YIG sphere can be formed. In the optical system, the nonlinear effect is also important to observe the nonreciprocal light transmission[15]. For the magnon–polariton system, this may be achieved by harnes-sing the nonlinear effect of magnons in the YIG sphere[40].

In conclusion, we have observed the exceptional point and spontaneous symmetry breaking in the cavity magnon–polariton system, where CPA is achieved in the unbroken-symmetry regime but not in the broken-symmetry regime. Meanwhile, the experi-mental results clearly display a phase transition of the system from the MIT to the weak-coupling regime. Our experiment demonstrates the inherent non-Hermitian nature of the cavity magnon–polaritons and offers a tunable macroscopic quantum platform for exploring the physics of non-Hermitian systems.

## Methods

**Device description.** For our 3D rectangular cavity, the two lowest-order modes TE101 and TE102 have the resonant frequencies around 8.16 and 10.03 GHz, respectively. In the experiment, we focus on the TE102 mode and use $\omega_c$ to denote the resonant frequency of this mode. Also, we use a highly polished YIG sphere with a diameter of about 0.32 mm, which is glued at one end of a thin wooden rod with a diameter of 1 mm and inserted into the rectangular cavity through a side hole with a diameter of 5 mm. The YIG sphere can be moved along the long edge of the cavity (i.e., the x direction) (Fig. 1a) and its location is tuned by varying the insertion depth of the rod using a position adjustment stage. Here the crystalline axis ⟨100⟩ of the YIG sample is parallel to the long edge of the cavity. Moreover, we apply a static magnetic field to the YIG sphere along the short edge of the cavity (i.e., the y direction), which is also parallel to the crystalline axis ⟨110⟩ of the YIG sphere.

**Measurement setup.** The measurement is performed at room temperature and the device, i.e., the cavity with a small YIG sphere embedded, is mounted between the two poles of an electromagnet (Supplementary Note 1). The static magnetic field is tuned by the electric current and a Gauss meter is used to monitor the applied magnetic field via a Hall probe. To measure the reflection under one feeding on port 1, the microwave signal is injected into the cavity via port 1, reflected by the device, and detected using a VNA via a 20 dB directional coupler. While to observe CPA under two feedings, the microwave signal (0 dBm) generated

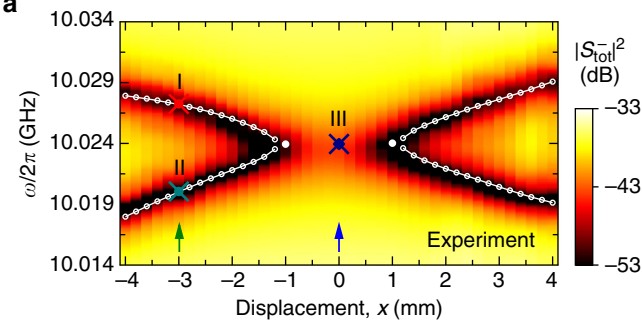

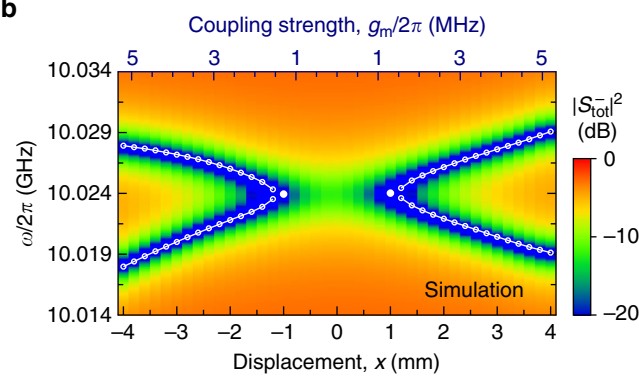

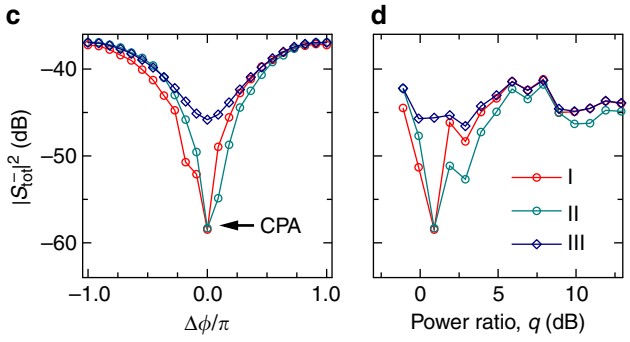

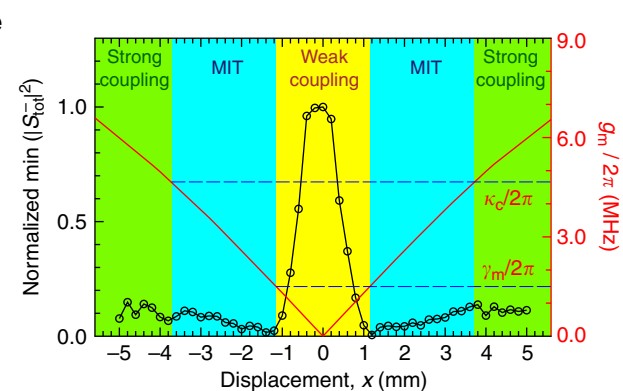

**Fig. 3** Observation of the exceptional point in cavity magnon–polaritons. **a** Measured total output spectrum vs. the displacement of the YIG sphere in the cavity. **b** Calculated total output spectrum corresponding to the measured results in **a**. Corresponding coupling strength is indicated according to the relation $g_m/2\pi = 1.3|x|$. In **a**, **b**, the open-circle curves are the eigenfrequencies calculated using Eq. (3). **c** Dependance of the total output spectrum on $\Delta\phi$. **d** Dependance of the total output spectrum on the power ratio. For the three curves (I, II, and III) in **c**, **d**, the corresponding locations in **a** are marked with crosses (red, green, and navy). **e** Phase transition between the magnetically induced transparency (MIT) regime and the weak-coupling regime. The black open-circle curve corresponds to the minima of the total output transmission vs. the displacement of the YIG sphere, where each value in the MIT and strong-coupling regions is the average value of the two minima of the two CPA branches at a given displacement

by the VNA is divided into two copies and injected into the cavity via the two ports of the cavity. The outputs from both ports are sent back to the VNA to measure the amplitude and phase responses of the device. In order to tune the power ratio and the phase difference between the feeding fields of ports 1 and 2, a variable attenuator and a phase shifter are placed in paths 1 and 2, respectively. The input link losses (associated with the signal from the VNA output to the cavity ports) from all components (cables, splitter, attenuator, phase shifter, directional couplers, etc.) are 34.2 and 35.1 dB for paths 1 and 2, respectively. By varying magnitude of the static magnetic field, the magnon frequency is tuned to approach to the resonant frequency of the cavity and then the device responses are recorded by sweeping the microwave signal frequency.

**Data availability**. The data that support the findings of this study are available from the corresponding author upon request.

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

## Acknowledgements

This work is supported by the National Key Research and Development Program of China (Grant No. 2016YFA0301200), the NSAF (Grant Nos. U1330201 and U1530401), the Most 973 Program of China (Grant No. 2014CB848700), and the Science Challenge Project (No. TZ2017003).

## Author contributions

D.Z. conceived the experiment. D.Z. and T.-F.L. designed and implemented the experimental setup. D.Z., X.-Q.L. and Y.-P.W. performed the experiment. D.Z. and T.-F.L. analyzed the data. J.Q.Y. and D.Z. worked out the theory. All authors contributed to the writing of the manuscript. J.Q.Y. and T.-F.L. supervised this work.

## Additional information

**Competing interests:** The authors declare no competing financial interests.

