## [Peer Review File · Nature Communications]

Reviewers' comments:

Reviewer #1 (Remarks to the Author):

This paper is a nice paper showing a simple tunable cavity, which shows a way of studying an exceptional point. I am not sure if it is up to "Nature Communication" standard, because I do not understand what their standard is. But what I can say is it is a very readable paper and the supplementary material gives a clear mathematical description. There is a paper that was recently published in PRB, which should be referenced.

PHYSICAL REVIEW B 95, 214411 (2017) Topological properties of a coupled spin-photon system induced by damping, Michael Harder,* Lihui Bai, Paul Hyde, and Can-Ming Hu

This paper is on the same topic, and I would expect this submitted paper to be suitable for PRB as well. So if the authors are not successful, I would encourage them to submit to PRB (and save money). However, I can say the physics is correct, the paper is well organized, and I would judge it suitable for publication in this Journal. But I have been overruled on my references before by this journal, so I will leave it to the editors to decide as usual.

Reviewer #2 (Remarks to the Author):

In the present work, the authors exploit the strong coupling regime between magnons and photons to obtain, dynamically balancing pumping and loss, stationary polariton states. They design a tunable cavity system to observe polaritonic coherent perfect absorption (CPA) and obtain evidence of an exceptional point. They finally propose their cavity system as a platform for exploring the physics of non-Hermitian systems.

In my opinion the paper is well-organized, the results are sound and there is good evidence for its conclusions. Therefore, I would like to suggest the acceptance of the paper as long as the authors could consider my following concerns.

1. In optical systems it has been shown (A. Mostafazadeh J. Phys. A: Math. Theor. 45 444024, 2012) that self-dual spectral singularities and CPA can be obtained without PT-symmetry. The authors should comment on similar possibilities for the cavity magnon-polariton scheme under study and clarify whether or not PT-symmetry is necessary to have exceptional points. In my opinion this is an important point since, although the manuscript is interesting for scientists in the specific field, the investigation of possible designs stemming from the absence of PT symmetry, could surely enlarge the interested audience.
2. Moreover, in the introduction, the authors claim that "the system of cavity photons coupled to magnons can inherently possess non-Hermiticity ". Do they mean that this is a peculiar characteristic of magnon-polaritons or of all polaritonic quasiparticles?
3. It could be useful to have a phase diagram to better identify the phase transition.

Reviewer #3 (Remarks to the Author):

The work by Zhang et al experimentally demonstrated a cavity magnon-polariton system can be used as an ideal platform for studying non-Hermitian physics. Hybrid systems using 3D microwave cavity and YIG spheres have been studied intensively recently because of their great potential in quantum information processing, and coherent coupling of YIG magnon with other systems such as microwave cavities and superconducting qubits have been demonstrated. In this work, the authors demonstrated polaritonic coherent perfect absorption and the phase transition at the

exceptional point. This brings the cavity magnon-polariton system into a new regime (non-Hermitian) for the first time, which deepens the understanding and enriches the study of such a versatile system. It also provides a new platform for studying non-Hermitian physics, enabling new methods and toolsets in exploring this important area of quantum physics.

In this manuscript, the experiment is well designed and the data is clearly presented, and the related theory analysis is thorough and clear. In general, it is a very nice piece of work. But to be published in Nature Communications, the authors have to address the following issues first – mostly related to the significance this work/system, plus some minor issues:

1. The authors should elaborate more on the significance of their work. Why is the magnon-polaritonic non-Hermiticity important and what are the potential applications? How is the magnon-polaritonic system unique compared with other already demonstrated systems (optical, atomic, etc.) in non-Hermitian physics? For example, magnons are intrinsically tunable and nonreciprocal, how would these properties contribute to the study of non-Hermitian physics? Besides, would the fact that the magnon-polaritonic system is a hybrid system brings new possibilities (compared with other non-Hermitian systems)?
2. The demonstrated system is described by a non-Hermitian PT-symmetric Hamiltonian. Could the authors comment about whether such a system can be used to observe PT-symmetry phenomena such as nonreciprocal light transmission?
3. Although the phase transition from weak coupling to MIT is evident in Fig. 3a, it is helpful to plot a linecut (the minima of $|S_{\text{tot}}|^2$ as a function of displacement) to better visualize the threshold behavior of the phase transition at the exceptional point.
4. In the caption of Fig 3d, the expression $q=k_1/k_2$ is a little confusing. Here there should be two variables, one is q , which is defined as the power ratio and can be tuned externally by the variable attenuator, while the other one is q_0 , which is fixed as k_1/k_2 and represents the specific power ratio required for the CPA condition.
5. The authors should carefully check the text and correct the typos, particularly in the mathematical expressions and variables. Here are a few examples: Line 1 below Figure 2, “ k_1 and k_1 ”. Line 7 on Page 5, left column, “... strong coupling ($g_m > k_c$) or MIT ($g_m > k_c$)”.

Reply to reviewer #1's comments

We would like to thank reviewer #1 for his/her comments and suggestions. Below we provide our reply.

> This paper is a nice paper showing a simple tunable cavity, which shows a way of studying an exceptional point.
> I am not sure if it is up to "Nature Communication" standard, because I do not understand what their standard is.
> But what I can say is it is a very readable paper and the supplementary material gives a clear mathematical
> description. There is a paper that was recently published in PRB, which should be referenced.

> PHYSICAL REVIEW B 95, 214411 (2017) Topological properties of a coupled spin-photon system induced by
> damping, Michael Harder,* Lihui Bai, Paul Hyde, and Can-Ming Hu

> This paper is on the same topic, and I would expect this submitted paper to be suitable for PRB as well. So if the
> authors are not successful, I would encourage them to submit to PRB (and save money). However, I can say the
> physics is correct, the paper is well organized, and I would judge it suitable for publication in this Journal. But
> I have been over ruled on my references before by this journal, so I will leave it to the editors to decide as usual.

Reply:

We thank reviewer #1 for his/her recommendation. In the revised manuscript, we have cited this PRB paper (mentioned by referee #1) as Ref. [36] in the 'Discussion' section.

Reply to reviewer #2's comments

First of all, we would like to thank reviewer #2 for his/her useful comments and suggestions. Below we respond to them one by one.

> In the present work, the authors exploit the strong coupling regime between magnons and photons to obtain,
> dynamically balancing pumping and loss, stationary polariton states.
> They design a tunable cavity system to observe polaritonic coherent perfect absorption (CPA) and obtain
> evidence of an exceptional point. They finally propose their cavity system as a platform for exploring the
> physics of non-Hermitian systems.
> In my opinion the paper is well-organized, the results are sound and there is good evidence for its conclusions.
> Therefore, I would like to suggest the acceptance of the paper as long as the authors could consider my
> following concerns.

Reply:

We thank reviewer #2 for this positive comment and also for his/her recommendation.

- > 1. In optical systems it has been shown (A. Mostafazadeh J. Phys. A: Math. Theor. 45 444024, 2012) that
- > self-dual spectral singularities and CPA can be obtained without PT-symmetry. The authors should comment
- > on similar possibilities for the cavity magnon-polariton scheme under study and clarify whether or not
- > PT-symmetry is necessary to have exceptional points.
- > In my opinion this is an important point since, although the manuscript is interesting for scientists in the
- > specific field, the investigation of possible designs stemming from the absence of PT symmetry,
- > could surely enlarge the interested audience.

Reply:

We also thank reviewer #2 for this useful comment. Following his/her suggestion, we have added the following paragraph (i.e., the second paragraph) in the 'Discussion' section:

“Recently, cavity magnon-polaritons are proposed to be used as hybrid quantum systems for quantum information processing [23,26], where the engineering of dissipations is a crucial part. In our experiment, we properly engineer the ports of cavity and the related decay rates to implement a cavity magnon-polariton system with the non-Hermitian PT-symmetric Hamiltonian. We show that the CPA can be utilized to identify the transition from the unbroken-symmetry regime to the broken symmetry regime at the exceptional points in this hybrid system. However, in optical systems, it has been shown [35] that self-dual spectral singularities and CPA can be obtained without PT symmetry. For the cavity magnon-polariton scheme, this should also be possible because the quantum description of the cavity magnon-polariton system under the low-lying excitation of magnons is analogous to an optical system. Moreover, as a new platform for non-Hermitian physics, the cavity magnon-polaritons possess the merits of both magnons and cavity photons. It will provide more degrees of freedom to investigate possible designs stemming from the absence of PT symmetry, including the observation of exceptional points in the non-Hermitian system without PT symmetry [35-37].”

- > 2. Moreover, in the introduction, the authors claim that “the system of cavity photons coupled to magnons can
- > inherently possess non-Hermiticity”. Do they mean that this is a peculiar characteristic of magnon-polaritons
- > or of all polaritonic quasiparticles?

Reply:

In the real system, the damping rate of magnons is fixed for a given sample. In order to have the cavity magnon-polaritons well described by a non-Hermitian Hamiltonian, we need to engineer the ports of cavity for inputting microwave photons and the related decay rates. In the revised manuscript, for more clarity, we have replaced the sentence “the system of cavity photons coupled to magnons can inherently possess non-Hermiticity” by the sentence below (see page 1):

“While the damping rate of magnons is fixed, by engineering the ports of cavity for inputting microwave photons and the related decay rates, the system of cavity photons coupled to magnons can be described by a non-Hermitian Hamiltonian.”

> 3. It could be useful to have a phase diagram to better identify the phase transition.

Reply:

We thank reviewer #2 for this suggestion. In the revised manuscript, we have added a new figure, namely Fig. 3e (as well as its caption), and two sentences in the first paragraph of the ‘Discussion’ section to better identify the phase transition at the exceptional point.

Reply to reviewer #3's comments

First of all, we would like to thank reviewer #3 for his/her useful comments and suggestions. Below we respond to them one by one.

> The work by Zhang et al experimentally demonstrated a cavity magnon-polariton system can be used as an ideal
> platform for studying non-Hermitian physics. Hybrid systems using 3D microwave cavity and YIG spheres have
> been studied intensively recently because of their great potential in quantum information processing, and
> coherent coupling of YIG magnon with other systems such as microwave cavities and superconducting qubits
> have been demonstrated. In this work, the authors demonstrated polaritonic coherent perfect absorption and
> the phase transition at the exceptional point. This brings the cavity magnon-polariton system into a new regime
> (non-Hermitian) for the first time, which deepens the understanding and enriches the study of such
> a versatile system. It also provides a new platform for studying non-Hermitian physics, enabling new
> methods and toolsets in exploring this important area of quantum physics.

> In this manuscript, the experiment is well designed and the data is clearly presented, and the related theory
> analysis is thorough and clear. In general, it is a very nice piece of work. But to be published in
> Nature Communications, the authors have to address the following issues first – mostly related to
> the significance this work/system, plus some minor issues:

Reply:

We thank reviewer #3 for this positive comment.

> 1. The authors should elaborate more on the significance of their work. Why is the magnon-polaritonic

- > non-Hermiticity important and what are the potential applications?
- > How is the magnon-polaritonic system unique compared with other already demonstrated systems (optical, atomic, etc.) in non-Hermitian physics? For example, magnons are intrinsically tunable and nonreciprocal,
- > how would these properties contribute to the study of non-Hermitian physics? Besides, would the fact that
- > the magnon-polaritonic system is a hybrid system brings new possibilities (compared with other non-Hermitian systems)?

Reply:

We thank reviewer #3 for this useful comment. Considering referee #3's suggestions, we have added some sentences in the second paragraph of the 'Discussion' section (see page 5) and also the following paragraph (i.e., the third paragraph) in the 'Discussion' section, so as to indicate the potential applications of the considered non-Hermitian magnon-polariton system:

"In addition, magnons are intrinsically tunable and nonreciprocal. In contrast to the optical and atomic systems, the hybrid system of cavity magnon-polaritons can provide new possibilities in non-Hermitian physics. For instance, pronounced nonreciprocity and asymmetry in the sideband signals generated by magnon-induced Brillouin scattering of light were observed in such a system [27]. When non-Hermiticity is considered, these properties may bring new features in the magnon-polariton system. Also, more small YIG spheres can be placed in a cavity. Tuning magnons in each YIG sphere, one can strongly couple magnons in these spheres to the cavity photons, either simultaneously or respectively. Then, one can implement quantum simulation of many-body bosonic models using this hybrid system and explore more interesting phenomena in non-Hermitian physics."

- > 2. The demonstrated system is described by a non-Hermitian PT-symmetric Hamiltonian. Could the authors
- > comment about whether such a system can be used to observe PT-symmetry phenomena such as
- > nonreciprocal light transmission?

Reply:

We also thank reviewer #3 for this useful comment. Following his/her suggestions, we have added the following paragraph (i.e., the fourth paragraph) in the 'Discussion' section (see page 5):

"Moreover, nonreciprocal light transmission was observed in an optical system with PT symmetry [15]. With the similarity to the optical system, it seems possible to demonstrate nonreciprocal light transmission in the cavity magnon-polariton system. For this, one can use one port of the cavity and then design a special port for the YIG sphere by, e.g., directly coupling the microwave field to the magnons via a coil [38]. Therefore, a round-trip signal transmission path between the cavity and the YIG sphere can be formed. In the optical system, the nonlinear effect is also important to observe the nonreciprocal light transmission [15]. For the magnon-polariton system, this may be achieved by harnessing the nonlinear effect of magnons in the YIG sphere [38]."

- > 3. Although the phase transition from weak coupling to MIT is evident in Fig. 3a, it is helpful to plot a linecut
- > (the minima of $|S_{tot}|^2$ as a function of displacement) to better visualize the threshold behavior of

> the phase transition at the exceptional point.

Reply:

We thank reviewer #3 for this helpful suggestion. In the revised manuscript, we have added a new figure, namely Fig. 3e (as well as its caption), and two sentences in the first paragraph of the 'Discussion' section (see pages 4 and 5), to better visualize the threshold behavior of the phase transition at the exceptional point.

> 4. In the caption of Fig 3d, the expression $q=k_1/k_2$ is a little confusing. Here there should be two variables,
> one is q , which is defined as the power ratio and can be tuned externally by the variable attenuator, while
> the other one is q_0 , which is fixed as k_1/k_2 and represents the specific power ratio required for
> the CPA condition.

Reply:

We fully agree with this comment. In our manuscript, q denotes the power ratio between two feeding fields, and q has to be set to a specific value of k_1/k_2 for the CPA condition. In the revised manuscript, we have replaced the expression ' $q=k_1/k_2$ ' with the phrase 'power ratio' in the caption of Fig. 3d, so as to eliminate the ambiguity.

> 5. The authors should carefully check the text and correct the typos, particularly in the mathematical
> expressions and variables. Here are a few examples: Line 1 below Figure 2, " k_1 and k_1 ".
> Line 7 on Page 5, left column, "... strong coupling ($g_m > k_c$) or MIT ($g_m > k_c$)".

Reply:

In the revised manuscript, the typos mentioned by referee #3 are corrected. Also, some sentences are polished.

REVIEWERS' COMMENTS:

Reviewer #2 (Remarks to the Author):

The authors have well addressed my previous concerns in the current manuscript. Therefore, I would like to suggest the acceptance of the paper.

Reviewer #3 (Remarks to the Author):

I am glad to see the authors have addressed all my comments. Particularly, I like the added paragraph in the "Discussion" section about the potential of the proposed system. Just one minor comment: when discussing the possibility of adding more YIG spheres in the system, there are a few closely related experimental work should be cited as a strong support, such as: "Magnon dark modes and gradient memory", Zhang et al, Nature Communications 6, 8914 (2015); "Cavity Mediated Manipulation of Distant Spin Currents Using a Cavity-Magnon-Polariton", Bai et al, Phys. Rev. Lett. 118, 217201 (2017). Other than that, I think the current version of the manuscript is in a very good shape. So I would recommend acceptance of this manuscript for publishing on Nature Communications.

Reply to reviewer #2's comments

> The authors have well addressed my previous concerns in the current manuscript. Therefore, I would like to
> suggest the acceptance of the paper.

Reply:

We thank reviewer #2 for his/her recommendation.

Reply to reviewer #3's comments

> I am glad to see the authors have addressed all my comments. Particularly, I like the added paragraph in the
> "Discussion" section about the potential of the proposed system. Just one minor comment: when discussing the
> possibility of adding more YIG spheres in the system, there are a few closely related experimental work
> should be cited as a strong support, such as: "Magnon dark modes and gradient memory", Zhang et al,
> Nature Communications 6, 8914 (2015); "Cavity Mediated Manipulation of Distant Spin Currents Using
> a Cavity-Magnon-Polariton", Bai et al, Phys. Rev. Lett. 118, 217201 (2017). Other than that, I think the
> current version of the manuscript is in a very good shape. So I would recommend acceptance of
> this manuscript for publishing on Nature Communications.

Reply:

We thank reviewer #3 for this positive comment and also for his/her recommendation. In the revised manuscript,
we have cited these two papers as Ref. [38,39] in the 'Discussion' section.